# Can We Completely Trust in Automated Software for Preoperative Planning of Shoulder Arthroplasty? Software Update May Modify Glenoid Version, Glenoid Inclination and Humeral Head Subluxation Values

**DOI:** 10.3390/jcm12072620

**Published:** 2023-03-30

**Authors:** Raffaele Garofalo, Alberto Fontanarosa, Alessandro Castagna, Nunzio Lassandro, Angelo Del Buono, Angelo De Crescenzo

**Affiliations:** 1Department of Orthopaedics and Traumatology, Ente Ecclesiastico Ospedale “F. Miulli”, Strada Prov. 127 Acquaviva–Santeramo Km. 4, Acquaviva delle Fonti, 70021 Bari, Italy; 2Department of Orthopaedics and Traumatology, Humanitas University Milano, Via Rita Levi Montalcini 4, Rozzano, 20090 Milano, Italy; 3Department of Orthopaedics and Traumatology, Ospedale Luigi Curto, Via Luigi Curto, 84035 Polla, Italy

**Keywords:** automated software, preoperative planning, shoulder arthroplasty, software update

## Abstract

Background: The purpose of this study was to evaluate the impact of software updating on measurements of the glenoid inclination and version, along with humeral head subluxation performed by an automated 3D planning program. The hypothesis was that the software update could significantly modify the values of the glenoid inclination and version, as well as of the humeral head subluxation. Methods: A comprehensive pool of 76 shoulder computed tomography (CT) scans of patients who underwent total shoulder arthroplasty (TSA) or reverse total shoulder arthroplasty (RTSA) were analyzed with the automated program Blueprint in 2018 and again in 2020 after a software update. Results: A statistically significant difference of 8.1 ± 8.2 and 5.4 ± 7.8 (mean difference of −2.8 ± 5.0, *p* < 0.001) was indeed reached when comparing the mean glenoid inclination achieved with Blueprint 2018 and Blueprint 2020, respectively. The glenoid version, as well as the humeral head subluxation evaluations, were not significantly different between the two software versions, with mean values being −9.4 ± 8.9 and −9.0 ± 7.4 and 60.1 ± 12.6 and 61.8 ± 12.0, respectively (*p* = 0.708 and *p* = 0.115, respectively). In 22% of CT scans, the software update determined a variation of the glenoid inclination of more than 5° or 10°. Conclusion: The present study shows the software update of an automated preoperative planning program may significantly modify the values of glenoid inclination. Even though without a significant difference, variations were also found for the glenoid version and humeral head subluxation. Accordingly, these results should further advise surgeons to carefully and critically evaluate data acquired with automated software.

## 1. Introduction

Total shoulder replacement represents a feasible and effective option for several degenerative shoulder conditions, with both anatomic (TSA) and reverse prosthesis design (RTSA) [1,2]. Results have improved over time in terms of patient satisfaction, pain relief and shoulder function. However, glenoid placement may strongly affect the long-term outcomes for potential glenoid loosening, instability and scapular notching [3,4,5,6,7,8,9]. Accordingly, broad knowledge of preoperative glenoid deformity is crucial to approaching any scenario for a correct implantation of the glenoid prosthetic component. 

Preoperative glenohumeral pathoanatomy has been traditionally evaluated with radiographs [10], but with computed tomography (CT) scans, the variability in glenoid morphology and its consequences for shoulder arthroplasty are more accurately studied [2,11]. Based on the CT scan potential, several 2D and 3D models were developed to preoperatively characterize the glenoid version, inclination and joint alignment. To address the variability of preoperative models with manual interference and improve measurement accuracy accordingly, three-dimensional (3D) automated software have been developed to plan glenohumeral arthroplasty [1,12]. This technology enables the surgeon to assess preoperatively the optimal positioning of the glenoid component for any clinical scenario and predict the inclination and version correction, as well as the backside contact. Moreover, these may provide a virtual reproduction of impingement-free range of motion and create patient-specific instrumentation (PSI) [3,11,13]. 

Accordingly, an automated software can help significantly in preoperative planning to reduce the inter- and intra-observed variability of manual methods. However, to improve glenoid parameters, the proprietary algorithm for the image is subjected to modification over time. Thus, a software update could result in variations of the inclination, version and humeral head subluxation. Nonetheless, there is no evidence thus far on the effect that software updating could have on preoperative glenoid measurements and then on implant positioning.

The purpose of this study was to evaluate the impact of software updating on measurements of the glenoid inclination and version, along with the humeral head subluxation, performed by an automated 3D planning program and then to statistically analyze the relevance of these variations. The hypothesis was that the software update could significantly modify the values of the glenoid inclination and version, as well as of the humeral head subluxation.

## 2. Materials and Methods

This study is a retrospective analysis of a pool of preoperative CT scans of patients who underwent TSA or RTSA from 2016 to 2018 in a single hospital center (Miulli Hospital, Acquaviva delle Fonti, Bari, Italy). The scans were analyzed with the automated software Blueprint (Blueprint, Stryker, Kalamazoo, MI, USA) in the same period when the patients were scheduled to undergo surgery in order to evaluate the glenoid version, glenoid inclination and humeral head subluxation. The software version was the same for this period, which was defined as Time Zero. We re-analyzed the same group of 76 CT scans in 2020 with the aim to assess if and how the software update of the automated 3D preoperative planning program could affect the glenoid assessment. All the CT scans were performed using a standardized protocol and were suitable to be studied with the updated program. In this way, data were recorded to obtain two groups, dubbed Time Zero Blueprint and Blueprint 2020. Retroversion was recorded as a negative value and superior inclination as positive, and subluxation was expressed as a percentage of the humeral head posterior to the scapular plane.

This study was approved by the IRB of the authors’ affiliated institutions. All patients signed the informed consent to be enrolled in the study. Most of the patients included in the study (70 out of 76) underwent RTSA because of a non-functional shoulder with a massive and irreparable rotator cuff tear (20 patients), post traumatic osteoarthritis (11 patients), rheumatoid arthritis (2 patients), rotator cuff arthropathy (28 patients) or primary glenohumeral arthritis (9 patients). TSA was performed in the remaining 6 patients (8%) suffering from primary glenohumeral arthritis with A1, A2, B1 glenoid morphology, according to Walch classification [1], and a good-quality rotator cuff. 

In all cases, patients preoperatively underwent radiographic study of both shoulders in anterior to posterior (AP), Neer and Bernageau view. All CT scan were done at the same radiological center, with the same machine and according to a standardized protocol [13]. CT scans were performed in all patients with a minimum of 1 mm slice thickness, starting a few slices above the AC joint and including the entire scapula. Based on 2D CT imaging, an expert musculoskeletal radiologist classified the glenoid morphology on the coronal and transversal planes according to Sirveaux [14] and Walch [15], respectively. 

The present study explores and compares two versions of the same automated preoperative planning software. The evolution of the former Glenosys software (Imascap, Brest, France), Blueprint software allows an automated definition of crucial anatomic parameters for a preoperative planning program, such as the glenoid version, glenoid inclination and humeral head posterior subluxation. A detailed description of the automated process of the scapula and humerus analysis and reference planes definition was described by Boileau et al. [2].

Briefly, this system uses image recognition technology to automatically isolate, using thousands of points within the scapula, a best-fit scapula plane, including the glenoid area, and a 3D model of the scapula. The typical curvature of the scapular blade may complicate the definition of the plane orientation, but the application of a cloud of point reduces the effect of the curvature on the plane definition. A similar approach is applied to define the best-fit sphere of the glenoid surface and of the humeral head. An averaged estimation of the best-fit sphere approaching both the paleogleloid and neoglenoid surface is used in cases with a biconcave glenoid. The glenoid version is measured as the angle between the scapular plane and the glenoid centerline projected on the transverse scapular plane. The glenoid inclination is assessed in reference to a transverse line that is calculated with points picked at the center of the Y shape of the trionum scapolae. As these points are arranged in a curved shape, the transverse axis is calculated as a straight line, with the least-square method applied to all the points. The humeral head subluxation is measured as the volume of the humeral head sphere posterior to the midcoronal plane of the glenoid sphere in reference to the volume of the whole humeral head. No manual subtraction of bone fragments is performed with this planning program. In addition, the automated software creates a 3D reconstruction of the scapula and humerus, on which it performs a trial of the potential implant types and sizes. In this way, the program predicts the backside contact, and then the need for augmentation may create patient-specific instrumentation and virtually reproduce an impingement-free range of motion.

### Statistical Analysis

Data are reported as the mean ± standard deviation or a number with percentage. Differences were tested with the paired sample t-test. A *p* value of 0.05 or less was considered statistically significant. In addition, differences between each paired values of the glenoid version and inclination were gathered in the following groups: less than 5° difference, 5°–10° difference and more than 10° difference. As defined in previous studies [12,16], this analysis was performed considering a gap of 5°, as it is regarded as the slightest clinically relevant value. All analyses were conducted using STATA software, version 16 (Stata-Corp LP, College Station, TX, USA).

## 3. Results

A comprehensive group of 76 patients (76 CT scans) were examined, with any CT scan analyzed in 2018 suitable to be processed by Blueprint 2020. No patient was excluded from reanalysis. The mean patient age was 73 ± 6 years (range from 54 to 82 years) for the 18 men and 58 women included. According to the classification by Walch et al. [15], there were 48 A1, 8 A2, 13 B1, 6 B2 and 1 C glenoid. The glenoid was classified, based on Sirveaux’s Classification [14], as E0 in 32 patients, E1 in 38, E2 in 2, E3 in 3 and E4 in 1. The demographic and glenoid morphology classifications are shown in Table 1. 

The mean glenoid version was −9.4 ± 8.9 and −9.0 ± 7.4 with Time Zero Blueprint and Blueprint 2020, respectively, and no significant differences were reported (mean difference = 0.4 ± 8.5, *p* = 0.708; Table 2). A statistically significant difference was indeed reached when comparing the mean glenoid inclination achieved with Time Zero Blueprint and Blueprint 2020, measured at 8.1 ± 8.2 and 5.4 ± 7.8, respectively (mean difference of −2.8 ± 5.0, *p* < 0.001; Table 2; Figure 1). 

The humeral head subluxation evaluations were not significantly different among the two software versions, with 60.1 ± 12.6 and 61.8 ± 12.0, respectively (mean difference = 1.8 ± 5.3, *p* = 0.115; Table 2). Figure 2 shows a Bland–Altman plot comparing the glenoid and joint alignment parameters with the two software versions. A difference >5° between the two software versions was found in 68 patients (89%), 5°–10° in 6 patients (8%) and >10° in 2 patients (3%). The difference was more evident for inclination measurements, where a difference >5° was found in 59 patients (78%), between 5° and 10° in 13 patients (17%) and >10° in 4 patients (5%). When the analysis was restricted to patients with Walch classification types A1 and A2, the differences in the glenoid inclination assessment were more remarkable (mean difference of −3.0 ± 5.5 in A1-A2 types and −2.0 ± 3.4 for B1-B2-B3-C types, with *p* < 0.001 and *p* = 0.017, respectively). 

## 4. Discussion

The main finding of this study is that an automated software update may significantly modify the glenoid inclination over time. Important variations were also observed in the glenoid version and humeral head subluxation, even though they did not differ significantly. Therefore, we assume that the data of preoperative planning for shoulder arthroplasty performed with 3D automated software must always be considered critically in terms of the consequences of updating the software.

Glenoid loosening represented one of the most common causes of revision, with important risks factors recognized as excessive glenoid retroversion for TSA, and for RTSA, height and superior inclination, or too much retroversion or medialization [16,17,18,19]. Scapular notching (inferior, posterior or anterior) and implant instability are other severe complications linked to an improper glenoid replacement [6,8]. Accordingly, an accurate preoperative assessment is helpful to evaluate glenoid deformity and bone loss and then plan appropriate intraoperative correction.

Preoperative planning for shoulder arthroplasty was initially performed with plain radiographs, but soon after, it was replaced with 2D CT scans for more detailed imaging of bone tissue [20]. As a matter of fact, the accuracy of glenoid evaluation with 2D CT scans was discovered to be impaired by the plane selection on which the measurements are made [16,21,22,23,24]. Then, multiple studies clearly demonstrated a better assessment of the glenoid version and inclination with 3D CT reconstruction [21,23,25]. The 3D automated software was developed to provide glenoid parameters, unimpaired by external interference, thereby avoiding inter- and intra-observer variability. Moreover, these automated platforms may direct surgeons and all component staff towards definitive implanted prosthesis, anticipate potential bone or metallic augmentation, and reduce intraoperative unexpected events and related avoidable distress of staff as much as possible. This time- and energy-sparing practice should ultimately improve the accuracy of the implant placement and long-term outcomes.

Although all preoperative planning methods are CT scan-based, different landmarks and reference planes may introduce significant variability in the results achieved [22,23]. Traditional 2D and 3D methods determine the scapular plane by the manual positioning of three scapular points (glenoid center, most inferior point and the intersection between the scapular spine and the medial border of the scapula) and the glenoid plane with three points, as well, or with the best-fit sphere approach [2,25,26,27,28]. Conversely, the automated software obtains the scapular plane by elaborating thousands of 3D points of the scapular body and the glenoid plane by an automated definition of the best-fit sphere on the glenoid surface [2].

The reliability of glenoid measurements with fully automated 3D-planning software was previously demonstrated in an in vitro study on a non-arthritic shoulder [11,29] and then confirmed by Boileau et al. in vivo for arthritic scapulae [2].

Conversely, other studies have raised concern about the benefit afforded by automated software for preoperative planning [1,12,16,30]. In 2018, Denard et al. carefully compared the automated software Blueprint with the semi-automated program VIP (Arthrex, Naples, FL, USA) in 63 patients undergoing primary shoulder arthroplasty. In 57% of the cases (36 out of total 63 patients), either the inclination or the version varied by 5° or more, and in nearly 25% of the cases, the inclination or version varied more than 10° [12].

Erickson et al. recently compared the glenoid version and inclination achieved with four software programs (Blueprint, VIP, GPS and Materialise) and five surgeons [1]. Among the software analyzed, Blueprint presented the lowest agreement with surgeons’ measurements, with 56% of the version and 65% of the inclination within 5° of surgeons’ values. The authors explained this lower agreement as being due to a different geometrical algorithm and different mathematical calculations utilized. Moreover, they showed that software were prone to overestimate the glenoid version and humeral head subluxation (meant as increased retroversion and posterior subluxation). The high agreement among surgeons’ measurements was indeed deeply skewed by the coronal and axial plane selection, which were previously defined by a single senior author.

These findings were confirmed by a previous article of Chalmers et al. in 2017, which compared values delivered by Blueprint software with those measured on corrected and uncorrected CT scans of B2-type glenoid deformities [16]. Although the means of the values were not extremely different, the automated software produced values 3.5° more retroverted than both the corrected and uncorrected CT images [16].

Similarly, in 2019, Shukla et al. compared automated Blueprint software with the manual measuring technique. They found statistically significant differences, even though the differences of the values between the two methods were within 4° [30]. Referring to surgeries performed between 2013 and 2016, on average, the values delivered by Blueprint showed the glenoid more retroverted and superior-inclined compared to the plan achieved with the manual technique [30].

To the best of our knowledge, however, the present study is the first showing criticism of the software update with regard to the measurement of preoperative values, particularly in terms of the inclination. Moreover, even though a significant difference and a specific trend were not reached, variations were also found for both the glenoid version and the humeral head subluxation, with a mean difference of 0.4 ± 8.5 and 0.8 ± 5.3, respectively.

The reason for the demonstrated difference between the two software versions remains somewhat unclear. The automated subtraction of the humeral head may play a major role in determining the difference found between Time Zero Blueprint and the 2020 edition. The Blueprint program automatically removes the humeral head and any bone fragments, without any manual assistance. As observed by Denard et al., the accuracy of Blueprint may be influenced by factors related to the joint space narrowing and humeral head osteophytes [12]. In severe osteoarthritis with osteophytes and significant joint space narrowing, the automated process may fail to completely remove the humeral head or remove glenoid segments, skewing the final glenoid evaluation. In their paper, Denard et al. found the presence of extra bone fragments or missing segments in 30.6% of the CT scans analyzed with Blueprint [12]. Interestingly, the difference between Blueprint and VIP programs in the glenoid version assessment varied by 5° or more, from 19.4% of all evaluated cases to 45.5% when the analysis was restricted to patients with bone subtraction abnormalities [12]. Moreover, another reason for variations in the glenoid inclination measurements could stem from the transverse axis determination. The transverse axis is calculated with points picked at the center of the Y shape. Since these points are aligned forming a curve, the transverse axis is calculated as a straight line, with the least-square method applied to all the points. In the 2018 analysis, only the points of the two lateral thirds were considered to calculate the mean transverse axis. Then, a modification was made in 2020, taking into consideration the entire curvature of the points till the trigonum, with potential variation in the transverse axis determination. As recently demonstrated [31], a revised orientation of the transverse axis can significantly modify the glenoid inclination measurement.

Several companies have developed methods of performing preoperative glenoid measurements. However, remarkable differences can be observed comparing several methods since different companies use different measurement techniques [32]. Moreover, as the present study further remarks, an advanced and sophisticated automated software program is in an ongoing process of implementation and refinement. The software update is, in fact, intended as an improvement of the technology, which helps the preoperative planning. Nonetheless, this update needs subsequent validation. At the same time, the introduction of automated programs has certainly removed the inter- and intra-surgeon variability of manual measurements. However, while 3D automated software can achieve more detailed information on glenohumeral pathoanatomy, and a software update should improve the program accuracy over time, it remains unclear whether this additional information provides significantly better clinical outcomes for the patients.

This study has several limitations. The number of CT scans analyzed is relatively small. The patients with Walch type B and C glenoid deformities, for instance, were particularly limited for evaluating the effect of the software update in cases of more severe glenoid deformity. Conversely, this limited sample may have caused underestimation of the variation afforded by the software update.

Another potential limitation may be represented by case selection. In the present study, we considered the CT scans of several shoulder conditions, including degenerative, inflammatory and post-traumatic conditions. In the last case, the history of a previous trauma could have resulted in bone fragments or severe glenoid deformities that may potentially mislead the automated software. This was indeed part of our goal, that is, to check and probe the software reliability, even with complex deformities.

The main strength of the study relies on the solid data achieved directly by the software, without any kind of external spoiling. No kind of human interpretation or data processing was possible or necessary since the platform directly extrapolated the glenoid version, glenoid inclination and humeral subluxation.

## 5. Conclusions

The present study shows that a software update of an automated preoperative planning program may significantly modify the values of the glenoid inclination. Even though not significantly different, variations were also found for the glenoid version and humeral head subluxation. Consequently, these results should further advise surgeons to carefully and critically evaluate data achieved with automated software.

## Figures and Tables

**Figure 1 jcm-12-02620-f001:**
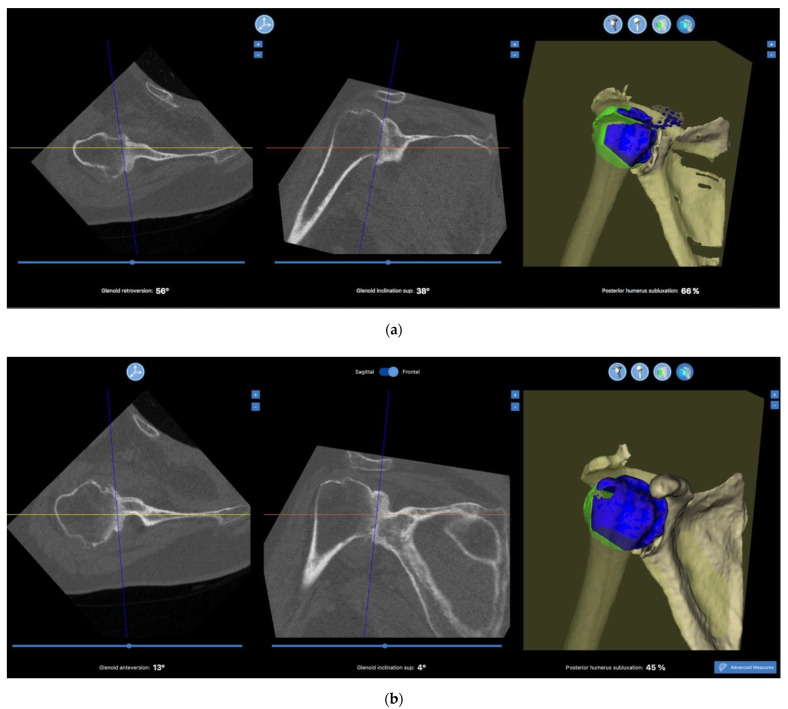
Two images depicting glenoid and joint alignment parameters of same patient with Time Zero Blueprint (**a**) and Blueprint 2020 (**b**).

**Figure 2 jcm-12-02620-f002:**
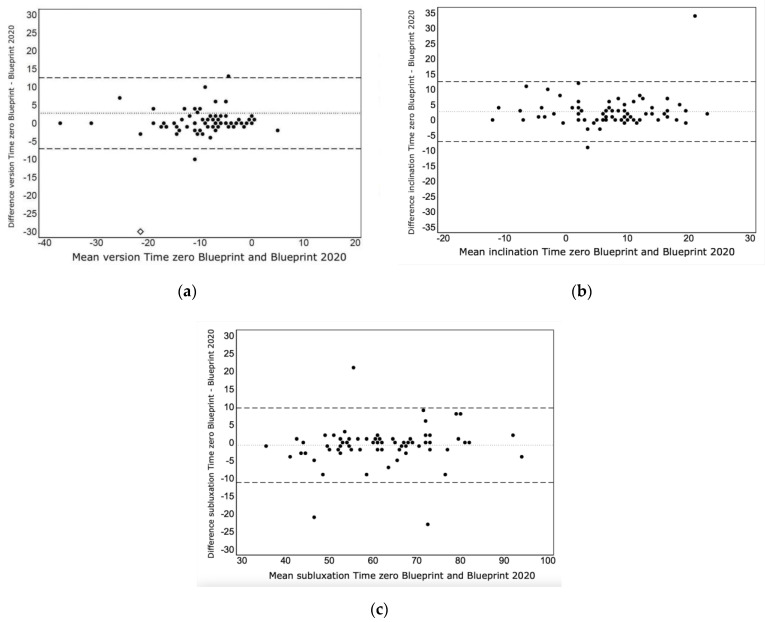
Bland-Altman plots showing variations achieved with software update for glenoid version (**a**), glenoid inclination (**b**) and humeral head subluxation (**c**).

**Table 1 jcm-12-02620-t001:** Characteristics of study population.

Patients (number)	76
Age (years)	73 ± 6
Males	18 (24%)
Females	58 (76%)
Side	
Right	48 (63%)
Left	28 (37%)
Walch modified classification	
A1	48 (63%)
A2	8 (11%)
B1	13 (17%)
B2	6 (8%)
C	1 (1%)
Sirveaux–Favard Classification	
E0	32 (42%)
E1	38 (50%)
E2	2 (3%)
E3	3 (4%)
E4	1 (1%)

**Table 2 jcm-12-02620-t002:** Glenoid and joint alignment measurements achieved with Time Zero Blueprint and Blueprint 2020. Values are represented as mean with standard deviation.

	Time Zero Blueprint	Blueprint 2020	Difference	*p* Value
Glenoid version (°)	−9.4 ± 8.9	−9.0 ± 7.4	+0.4 ± 8.5	0.708
Glenoid inclination (°)	8.1 ± 8.2	5.4 ± 7.8	−2.8 ± 5.0	<0.001
Humeral head subluxation (%)	60.1 ± 12.6	61.8 ± 12.0	+1.8 ± 5.3	0.115

## Data Availability

Data sharing not applicable.

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
