# Peer review of "Can We Completely Trust in Automated Software for Preoperative Planning of Shoulder Arthroplasty? Software Update May Modify Glenoid Version, Glenoid Inclination and Humeral Head Subluxation Values"

_jcm, 2023, doi:10.3390/jcm12072620_

Round 1

Reviewer 1 Report

Review JCM

The study aimed to evaluate the differences in accuracy using an automated software system to determine glenoid deformity in preoperative planning of shoulder arthroplasty using Blueprint software before and after software update. The study was well-designed, with a small but appropriate sample size and clear methodology. The results were also well-presented and easy to understand. Please consider the following recommendations below when making revisions:

Introduction: 

Overall sufficient with adequate references. 

Line 42: what do you mean with “glenoid management”? Please rephrase

Line 54: Please add “the” before glenoid component

Lines 58-63: This paragraph could be confusing to the reader. Maybe you could be more specific. We know that technologies changes quickly, but was there a Blueprint software update between 2018 and 2020 that supposedly improved automated measurements? If so, I would state that.

Methods:

Appropriate

Line 80: Remove “one”

Results: 

Appropriate

Discussion:

Line 216: Replace “as” by “that”

Lines: 238 and 240: Please add “the” before humeral head

Line 279: Replace “as” by “that”

Figures: 

Relevant and detailed captions. Images able to be interpreted easily.

References:

Appropriate

Overall Recommendations:

I recommend publishing the article due to its interesting findings and strong methodology. The results of the study seem to add valuable new information to the field, and the methods used to gather and analyze the data are well-designed and appropriate.

Author Response

Line 42: what do you mean with “glenoid management”? Please rephrase

Response: sentence modified

Line 54: Please add “the” before glenoid component

Response: words added.

Lines 58-63: This paragraph could be confusing to the reader. Maybe you could be more specific. We know that technologies changes quickly, but was there a Blueprint software update between 2018 and 2020 that supposedly improved automated measurements? If so, I would state that.

Response: thank you for the comment. The paragraph has been modified.

Line 80: Remove “one”

Response: removed

Line 216: Replace “as” by “that”

Response: modified

Lines: 238 and 240: Please add “the” before humeral head

Response: added

Line 279: Replace “as” by “that”

Response: modified

Reviewer 2 Report

Authors report differences between measurements made with 2 different versions (2018 and 2020) of the same planning software (Blueprint – STRYKER), and advocates this should call into question the reliability of the automated software.

The article is well written and very clear, and respect the normal structure of a scientific article.

I would have 3 main comments relative to this work.

The first is methodological.

The results illustrate clearly the error that has been made here.

I will take the example of the version : for 89% of the shoulders, an error of more than 5° was found, but the mean calculated error is only 0.4° (-9° and -9.4°) and is not significant. This is not possible.

Obviously, this is explained by the use of “signed” (+ or -) values to calculate the means, which is a frequent error in those type of studies.

An error should be an absolute value.

If you make an error of 5° more retroversion (-5°) or an error of 5° less retroversion (+5°)… this is always an error of 5°. If positive and negative errors are mixed in the calculation of the means, the “+” compensate the “–“, and you find no difference.

Consider 1 group of 2 patients. In one patient, you make an error of 5°. In the other, you make and error of -5°. If you calculate the mean, then you find 0° (no error…)… whereas you made a 5° error for each patient, and actually, the mean error is 5° (absolute value).

If authors wants to evaluate the error made between the 2 versions of the software, they have to record (for version, inclination and subluxation) the error made (no matter the sign + or -) and use absolute values to calculate the mean.

This has to be verified, as we have no access to the raw data, but it seems that values presented in this study have to be revised.

2) The second is about the explanation of the observed differences.

Authors make suppositions about the reasons explaining the observed differences “the reason of the showed difference between the 2 software remains unclear” L.237 – 251.

Actually, the reason explaining the differences is very clear, and authors would have probably obtained an answer with a simple discussion with the developpers and surgeons working on it.

The explanation is about the transverse axis determination. The transverse axis is calculated with points picked at the center of the Y shape. The alignement of these points is curved, and the transverse axis is a straight line calculated with the less-square method applied to all these points. In the 2018 version, only the points of the 2 lateral third were considered to calculate the mean transverse axis, considering this was enough. In the 2020 version, a modification has been made to take into consideration the entire curvature of the points til the trigonum, and this modified significantly the orientation of the transverse axis used as a reference for the calculations. But tTransverse axis is a very important parameter for the calculation of inclination (calculation of version and subluxation are less dependant) and this probably explains the observed differences. This has been demonstrated in a paper by MO GAUCI et al. (PMID : 36294372) which could be cited here.

3) The third is about the message.

Observed differences here are not explained by a mysterious bug inside the automated software… but is explained by scientifically supported decisions to modify the algorithm, in order to improve the measured values for version, inclination and subluxation.

This is improvement. Not necessarily a problem, or an error.

It is frequent that new versions bring modifications compared to the old one, in order to improve the functionality.

The message of the paper seems to discredit automated software, just because of this modification observed between 2 versions… which was actually an optimization, like there could be other in the future… Automated software are the more reliable option among 3D planning solution to date, producing identical measurements for a same patient in different attempts (the sentence “modify glenoid inclination over time” is confusing L169).

Manual software are time consuming, and subject to major error relative to inter or intra-observer variations, especially for the position of the 3 points defining the scapular plane (Jacquot et al. 36199507). With manual software, the variation is not a matter of software version, but a matter of defining the correct referential for each patient, and also for one only patient, if there is different attempts to measure. In the different publications comparing 2 or more software, the differences observed are usually condemning automated software, considering that manual 3D software, or even free measurements on CT-scan slices, would be the gold standard. The fact is that, at least, automated software do always the same job. This is only a problem of referential, which can be modified, optimized… With manual software, the referential has always been the same (3 points of Kwon et al.), but the way this referential is determined for each patient is subject to huge variations (inter- and intra-) and this could be the main problem, explaining the observed differences (assuming that the calculation of means have been made correctly). The discussion should consider this point of view.

Finally, a few comments about some details in the text :

L81 : humeral head subluxation is expressed as a percentage of the humeral head posterior to scapular plane (not scapular axis).

L98 : authors should precise the exact version used. Ex : Blueprint 2018 (v x.xxx.xx)

L114 : the transverse axis is not calculated exactly with the glenoid fossa, but with points picked at the center of the Y shape of the scapula. In the 2018 version, only the points of the 2 lateral thirds were considered to calculate the mean transverse axis. In the last versions (2020), the points were recorded on the entire scapula body until the trigonum.

As a conclusion, the idea is interesting, and the paper is well constructed, but the technical aspects are not well known by the authors, the data should be analyzed differently (absolute values to calculate mean errors), and the final message should be more weighted regarding the potential discredit of automated software.

Author Response

The first is methodological.

Response: I sincerely disagree with your comment, and I don’t understand your absoluteness in saying that it is “clearly” an error. I have checked again the statistical analysis with our statistician, and we answer that no error has been made. The comparison was performed within patients on the basis of difference between measurements. In absence of gold standard, that could be considered as a reference value to estimate the magnitude of deviation in absolute value, we simply estimated the mean of difference and tested its significance (Student t test for paired data) to quantify the probability of results due random sample selection. On the other hand, we have reported Bland and Altman plot to graphically display deviation (above and below) from the mean of every pair of measurements for each single patient.

If i don’t consider the sign “+” or “-” in the analysis, the difference we would find would be greater and with a more likely significant difference. Moreover, no information on a potential influence towards a more/less inclined or more/less retroverted glenoid due to the software update could be achieved.

2) The second is about the explanation of the observed differences.

Response: The discussion has been completed according the suggestion of the reviewer.

3) The third is about the message.

Response: We absolutely know that the modification achieved with software update aims to improve the correct definition of glenoid version and inclination, and humeral head subluxation. In fact, our study does not want to miscredit the automated software, but just warn against an uncritical use of these software. We sincerely and routinely use this software, but a manual measurement of the glenoid parameters is usually performed as well. However, we totally agree with the reviewer about the different referential as the root of inter- and intra- observer variability as explained at the lines 249-250 (“Although all preoperative planning methods are CT-scan based, different landmarks and reference planes may introduce significant variability in results achieved”). We agree obviously with the reviewer about the more constant values achieved by the automated software and the more time-consuming way to plan the surgery. However, if an automated software provides constant values removing the inter- and intra-observer variability, it’s at the same time true that update or improvement in the algorithm used can change the values achieved. Thus, the values we can achieve today could be different in the future, as it is happened in the last years. Then, they are constant but can change over time, because of an improvement process. However, this was not the aim of our work. I want to emphasize that it is absolutely not intended only for the BluePrint but for all available software. They can and are remarkably helping all of us, but we should always take care of an automated machine. In conclusion, we have never talked about a “bug” of the automated software as you have begun your third comment about the message. We have talked about a software “update” as clearly manifested starting from the title. This does not mean is a synonym neither of “error” nor of “problem”. If there is, instead, any issue with mention to the specific software, we have no problem to remove any reference to the specific software.

L81: humeral head subluxation is expressed as a percentage of the humeral head posterior to scapular plane (not scapular axis).

RESPONSE: modification has been made

L98 : authors should precise the exact version used. Ex : Blueprint 2018 (v x.xxx.xx)

RESPONSE: we have not sure information about the specific software 2018 version. Accordingly, no mention has been made.

L114 : the transverse axis is not calculated exactly with the glenoid fossa, but with points picked at the center of the Y shape of the scapula. In the 2018 version, only the points of the 2 lateral thirds were considered to calculate the mean transverse axis. In the last versions (2020), the points were recorded on the entire scapula body until the trigonum.

RESPONSE: modification has been made.

Reviewer 3 Report

Reviewer Comments:

General Comments:

All – there are some grammatical errors present throughout that should be addressed. These include improper pluralization of words and tense errors, among others.

Introduction:

Line 44 – what do you mean by “glenoid replacement may strongly affect long-term "outcomes for potential glenoid loosening, instability, and scapular notching [3-9].”? Did you mean glenoid placement?

Lines 51-52 – this line is unclear.

Lines 52-57 – I think you need to clarify what part of the ‘automated’ software you are talking about here. Are you talking about automated segmentation, anatomic measurement generation, or both? Blueprint is terms as a ‘Surgeon Controlled 3D Planning Software’, so it is not being marketed as an automated system apart from the automatic segmentation and anatomic plan generation, which are taken into consideration by the clinician as part of the overall surgical plan.

Lines 60-69 – I would agree that updates will alter the algorithms that automatically segment the CT scan, as well as generate the planes and coordinate systems used to calculate the output parameters.

Methods:

Line 79 – which hospital center?

Line 82 – the first analysis was performed all with the same version? This should be confirmed in the text as it currently just states between 2016 and 2018.

Line 124 – is there a more appropriate anatomic term that ‘Y shape’? Also, perhaps a simple figure would make this text easier for the reader to understand?

Results:

Table 2 – again, I don’t like ‘time zero blueprint’ as it is unclear whether it was all one version. If it was, state it like you have stated ‘blueprint 2020’. This table also needs units defined.

Discussion:

Line 212 – the reported glenoid inclination changed with the new version. Was it improved or was it worsened? I would assume that updates would serve to make the measurements more correct. Why did you not calculate the corresponding parameters manually with multiple observers and determine not only if the measurements changed, but if they were more closely aligned with manually selected values? This would be of more clinical importance.

Furthermore, the variation amongst observers would also be interesting to compare to the changes in the software; perhaps intra-observer differences are larger than the software changes (as these measurements are a bit objective).

Lines 215-217 – I would think that most users of the software are also interpreting the presented results as such and are comparing them with their own judgement.

Lines 218-223 – any resulting error in glenoid component placement is likely a combination of surgical variability, the inherent error in guide or surgical navigation systems, as well as the objective selection of the ‘best’ position of implantation. The pre-operative assessment is just one piece of a successful surgical procedure. I would recommend softening your statement that the preoperative assessment is mandatory…it is an important part of the entire process.

Line 233 – what is meant by ‘all components’ staff’?

Lines 276-278 – I think the main criticism I have of this work is that you do not know what changed between versions; for example, did the segmenting algorithm change? Or did the means by which the planes were selected from the geometry change? We have no indication as to whether these differences would occur once again if you repeated your work with the current version of the software, nor if the differences were improvements over the old version.

Lines 279-280 – these differences are so small (on average) and for a paired analysis any systematic difference would have surely been detected. It appears that the variation is in both directions. If so, this could be one of your conclusions here?

Line 295 – again, the ‘Y point’ could likely be better described as the ‘trigonum scapulae’?

Lines 298-301 – above you say the reasons are unclear, but here you present a valid point. Perhaps alter your wording above and start with this explanation?

Line 311 – do these software packages require ‘validation’ by the governing medial device bodies? What would ‘validated’ mean – accurate to what? The average of a series of clinician observers? This is an important point, and I would hope that all users of these software packages are award of the specific limitations of their inherent algorithms.

Line 317 – why do you think the scan number is small? What would be a good number to include? Did you perform a power analysis? How many in each group would you need to compare healthy to degenerate scapulae?

Conclusion:

General – agree with what is presented here. However, what is the clinical significance of this work? I cannot think of a case where a clinician would pre-plan a surgery, and then wait two years and revisit the case. They would have certainly forgotten the plan and would be a ‘new case’ for them. I also think its fairly well understood in the clinical community that the provided measurements are just estimates, and all clinicians take these into consideration with regards to their pre-operative planning. This is exacerbated by your study population which, as you admit, is laden with what are to be expected to be difficult cases with degeneration; something which is likely to make it harder for both clinicians and software packages to determine anatomic planes and measurements.

Author Response

Responses to Reviewer 3

General Comments:

All – there are some grammatical errors present throughout that should be addressed. These include improper pluralization of words and tense errors, among others.

Response: thank you, I have checked and corrected errors.

Introduction:

Line 44 – what do you mean by “glenoid replacement may strongly affect long-term "outcomes for potential glenoid loosening, instability, and scapular notching [3-9].”? Did you mean glenoid placement?

Response: yes exactly, I have changed as you suggested.

Lines 51-52 – this line is unclear.

Response: I have modified the sentence to make it clearer.

Lines 52-57 – I think you need to clarify what part of the ‘automated’ software you are talking about here. Are you talking about automated segmentation, anatomic measurement generation, or both? Blueprint is terms as a ‘Surgeon Controlled 3D Planning Software’, so it is not being marketed as an automated system apart from the automatic segmentation and anatomic plan generation, which are taken into consideration by the clinician as part of the overall surgical plan.

Response: i mean that Blueprint is an automated software in determining glenoid parameters, since differently by other semiautomated software, the software does not allow the surgeon to define plans or any others anatomical landmarks. Of course, the surgical planning is manual, but this is not the object of our work.

Lines 60-69 – I would agree that updates will alter the algorithms that automatically segment the CT scan, as well as generate the planes and coordinate systems used to calculate the output parameters.

Response: I agree too.

Methods:

Line 79 – which hospital center?

Response: I agree, I have modified the sentence.

Line 82 – the first analysis was performed all with the same version? This should be confirmed in the text as it currently just states between 2016 and 2018.

Response: yes, it is. The same version was used in the period between 2016 and 2018. The sentence has been modified to specify this point. Thank you for the suggestion.

Line 124 – is there a more appropriate anatomic term that ‘Y shape’? Also, perhaps a simple figure would make this text easier for the reader to understand?

Response: thank you for the suggestion, I have clarified the anatomic term as you suggest.

Results:

Table 2 – again, I don’t like ‘time zero blueprint’ as it is unclear whether it was all one version. If it was, state it like you have stated ‘blueprint 2020’. This table also needs units defined.

Response: the definition of “time zero Blueprint” has been clarified previously in the “Materials and Method”. The units have been clarified and added to the table. Thank you for the suggestion.

Discussion:

Line 212 – the reported glenoid inclination changed with the new version. Was it improved or was it worsened? I would assume that updates would serve to make the measurements more correct. Why did you not calculate the corresponding parameters manually with multiple observers and determine not only if the measurements changed, but if they were more closely aligned with manually selected values? This would be of more clinical importance.

Response: it was not the aim of our work. It is not possible to assert if the new values are better or worse. The aim was to evaluate the impact of software updating on measurements of glenoid inclination and version along with humeral head subluxation performed by an automated 3D planning program. Consequently, these results should further advise surgeons to carefully and critically evaluate data achieved with automated software.

Furthermore, the variation amongst observers would also be interesting to compare to the changes in the software; perhaps intra-observer differences are larger than the software changes (as these measurements are a bit objective).

Response: thank you, as discussed in the previous point, it was not our primary aim.

Lines 215-217 – I would think that most users of the software are also interpreting the presented results as such and are comparing them with their own judgement.

Response: it is exactly our aim.

Lines 218-223 – any resulting error in glenoid component placement is likely a combination of surgical variability, the inherent error in guide or surgical navigation systems, as well as the objective selection of the ‘best’ position of implantation. The pre-operative assessment is just one piece of a successful surgical procedure. I would recommend softening your statement that the preoperative assessment is mandatory…it is an important part of the entire process.

Response: yes, I agree with you. It is just a piece for a successful surgical procedure. I think it is crucial to preoperatively plan and know as best as you can which are the glenoid version and inclination and accordingly how to perform the surgery. However, I have lightened the statement as you suggest.

Line 233 – what is meant by ‘all components’ staff’?

Response: I mean surgeons and nurses.

Lines 276-278 – I think the main criticism I have of this work is that you do not know what changed between versions; for example, did the segmenting algorithm change? Or did the means by which the planes were selected from the geometry change? We have no indication as to whether these differences would occur once again if you repeated your work with the current version of the software, nor if the differences were improvements over the old version.

Response: the main object of our work is to show how a helpful tool (we routinely employ this software for our daily work) can nonetheless change the values we daily use to plan shoulder arthroplasty. The modification on inclination has been statistically significant. The modifications observed with new software version are surely intended by all of us of an improvement of the software, even though there is not a defined gold standard for preoperative glenoid evaluation. These modifications surely are the results of a new way to identify the anatomic landmarks on which measure these angles. Thus, this means that the perfect evaluation still does not exist, then as a consequence of our study we advise surgeons to carefully and critically evaluate data achieved with automated software.

Lines 279-280 – these differences are so small (on average) and for a paired analysis any systematic difference would have surely been detected. It appears that the variation is in both directions. If so, this could be one of your conclusions here?

Response: no, the variations is not in both directions. There is a trend in each variation, but the difference reached a statistically significance for the glenoid inclination.

Line 295 – again, the ‘Y point’ could likely be better described as the ‘trigonum scapulae’?

Response: thank you for the suggestion, I have clarified the anatomic term as you suggest.

Lines 298-301 – above you say the reasons are unclear, but here you present a valid point. Perhaps alter your wording above and start with this explanation?

Response: I say it is some somewhat unclear; then I say some of potential explanations for these differences.

Line 311 – do these software packages require ‘validation’ by the governing medial device bodies? What would ‘validated’ mean – accurate to what? The average of a series of clinician observers? This is an important point, and I would hope that all users of these software packages are award of the specific limitations of their inherent algorithms.

Response: when I say that the software needs a validation, I mean that they should be compared with other existing software or manual evaluation and then always critically analyze these results.

Line 317 – why do you think the scan number is small? What would be a good number to include? Did you perform a power analysis? How many in each group would you need to compare healthy to degenerate scapulae?

Response: the number is however enough to draw some important conclusions, maybe a bigger sample could give us even more information. No power analysis was performed.

Conclusion:

General – agree with what is presented here. However, what is the clinical significance of this work? I cannot think of a case where a clinician would pre-plan a surgery, and then wait two years and revisit the case. They would have certainly forgotten the plan and would be a ‘new case’ for them. I also think its fairly well understood in the clinical community that the provided measurements are just estimates, and all clinicians take these into consideration with regards to their pre-operative planning. This is exacerbated by your study population which, as you admit, is laden with what are to be expected to be difficult cases with degeneration; something which is likely to make it harder for both clinicians and software packages to determine anatomic planes and measurements.

Response: the message our work would show is that an automated software, which is probably the most used for the preoperative planning can give us measurements so different by the time with software update. Thus, the same “patient” (or two patients with the same glenoid and scapula shape and features) could be also managed potentially differently if the values are close to some important cutoff for the choice of glenoid components for example with augment or wedge. About your last point, cases with severe arthrosis and bone degeneration are also the most difficult to treat and theoretically the most frequent patients.

Round 2

Reviewer 2 Report

I read the answer of the authors. Ok with most of the remarks and modifications.   I would like to draw your attention on the methodological point, regarding the use of « signed » values for the calculation of mean errors.   Despite what the authors advocates, and the fact they « discussed this point » with their statistician… this is not correct, and a very common error that is made in many papers I had to review with this kind of calculation. There is no need to ask a statistician, this is only logic.   Here is a clear, and easy to understand, example to illustrate the problem : Let's consider a population of only 2 patients. For patient 1 the software make an error of 5° in anteversion (+5°). For patient 2 the software make an error of 5° in retroversion (-5°). In fact, the mean error is 5°, because in each case, the error that have been made is 5°, not matter this is an excessive anteversion or retroversion. But if you use « signed » values, making an average of -5° and +5°, you find a mean error of 0°, and conclude the precision is perfect… what is actually wrong. That is what I think the authors did (even if I dont have acess to the raw data) And this should be considered before accepting the publication. The fact that the sign is important to analyse if the error tend to be in ante- or retroversion is actually true, I agree with that, and the Bland Altman plot are useful to display these data. But when you calculate the mean error, the you have to use absolute value.

Author Response

Again, i disagree with your comment. In the former response, I have reported the comments of our statistician. I would show you again our point of view.

If i don’t consider the sign “+” or “-” in the analysis, the difference we would find would be greater and with a more likely significant difference. Moreover, no information on a potential influence towards a more/less inclined or more/less retroverted glenoid due to the software update could be achieved. Our aim is not to quantify the absolute value, it's more important to eventually identify a trend in the modified measurements achieved comparing the two groups. If I take off the sign from the statistic, we would not know if there is a trend towards the update is going. For this reason, we want to maintain the sign "+" or "-". I agree with our statistician. 

In our opinion, this makes more sense. This is in our opinion more as you say logical.

Thanks.